# LLM2Features: Large Language Models in Interpretable Feature Generation for AutoML with Tabular Data

## Abstract

Automatic Machine Learning (AutoML) is the popular supervised learning approach for tabular data. One of its key components is generating the most suitable features given the available training dataset. To overcome the disadvantages of existing automatic feature generation techniques, such as lack of generality and interpretability, we propose the novel approach, **LLM2Features**. It uses LLMs (Large Language Models) to generate meaningful features using automatically collected statistics about the dataset without explicitly describing the data, making it ideal for implementing in AutoML frameworks. In particular, we introduce the LLM-based critic that additionally verifies the presence of syntax or logical errors. The experimental study demonstrates the benefits of the proposed LLM2Features approach in accuracy and training time compared to the state-of-the-art feature generation tools.

## 1 Introduction

Nowadays, AutoML is widely used for training machine learning models on tabular (structured) data (Erickson et al., 2020; Fakoor et al., 2020; Li et al., 2021) as they allow to achieve high-quality results in several lines of code without the need to be an expert in choosing algorithms and their hyperparameters. One of the challenging steps in AutoML is the automated feature engineering (Mumuni & Mumuni, 2024) that can generate the most informative features for concrete task (Luo et al., 2019; Silva & Silva, 2023).

Existing feature generation methods have several disadvantages, namely, the need to input additional data from the human (Kanter & Veeramachaneni, 2015; Hollmann et al., 2024) or the impossibility of enriching the data sufficiently well without losing the interpretability of generated features (Zhang et al., 2023; Horn et al., 2020; Li et al., 2022). Hence, they may not be suitable for practical applications when data analysts have already designed a list of valuable and interpretable features. As a result, the most popular AutoML frameworks (Feurer et al., 2020; LeDell & Poirier, 2020; Vakhrushev et al., 2021) use existing feature generation frameworks (Kanter & Veeramachaneni, 2015) or traditional data pre-processing (Qi et al., 2023) for categorical variables, correct type conversion, etc.

This paper studies LLM (Large Language Model)-based automated feature generation techniques for the AutoML model that fits the generated features (Han et al., 2024). LLMs have been trained on a much larger amount of data including feature generation code, and have a good representation of the world they can use when generating features. The first successful application of LLM is the CAAFE (Context-Aware Automated Feature Engineering) framework (Hollmann et al., 2024). Unfortunately, it requires a detailed data description, so it cannot be implemented in typical AutoML solutions without human interaction. Moreover, the generated features sometimes lack meaningfulness and interpretability. Moreover, it is even possible that generted features contains mistakes of logical errors.

In this paper, we propose to use only the provided dataset itself without the need for any additional information. In particular, our **main contribution** is the novel approach, LLM2Features, which automatically extracts essential statistics from the dataset and feeds them into the prompt for feature generation using high-quality LLM, such as GPT-4o or GPT-o1. It is experimentally shown that

LLM-based feature generation for popular LightAutoML framework (Vakhrushev et al., 2021) has **much better quality metrics and human interpretability** of the features when compared to traditional feature engineering frameworks. Therefore, the proposed method can be used not only for generating interpretable features for AutopML but also for introductory **exploration of data in an unknown domain** to the analyst.

## 2 RELATED WORKS

### 2.1 PROBLEM STATEMENT

Given a dataset $D = \{(\mathbf{x}_1, y_1), (\mathbf{x}_2, y_2), \ldots, (\mathbf{x}_n, y_n)\}$, where $\mathbf{x}_i \in \mathbb{R}^m$ is an $m$-dimensional input vector, and $y_i$ is the corresponding target variable, the goal of feature generation is to **find a function** $\mathbf{z} = f(\mathbf{x})$, where $\mathbf{z}$ is a new $k$-dimensional feature vector. The objective is to **maximize the predictive performance of a mode**l $M$ trained on the new feature space: $\max P(M(D'))$, where $D' = \{(\mathbf{z}_1, y_1), (\mathbf{z}_2, y_2), \ldots, (\mathbf{z}_n, y_n)\}$ and $P$ is a performance metric (e.g., accuracy, ROC-AUC, RMSE). An additional requirement to automated feature generation is to minimize the level of **Human Involvement** (**H.I.**). In this paper, we use three different values for this metric:

- **0** is just to load the data (pd.DataFrame (Wes McKinney, 2010)). The best suitable method for use with AutoML

- **1** is to describe the data with free-form text (where the data comes from, what the nature of it is) to the prompt

- **2** is to preprocess the features with code (fill in the omissions, remove anomalies, cast the types (e.g., featuretools requirements)

### 2.2 AUTOMATIC FEATURE GENERATION

There exist two types of feature generation techniques, which 1) maximize the quality metrics by arbitrary transformations of features leading to the lack of interpretability (Bosch et al., 2021), or 2) generate logical, interpretable features by using knowledge of the world and data (Gosiewska et al., 2021). Among the first type of techniques, it is necessary to mention AutoFeat (Horn et al., 2020) feature generation by repeating various operations on one or a pair of features and using a built-in selector to select only helpful features. The OpenFE (Zhang et al., 2023) follows a similar principle but makes it faster with a specially developed boosting selector for a deeper understanding of the feature importance to the model in a further generation. The second technique that brings new information into the data is the featuretools library (Kanter & Veeramachaneni, 2015), which generates new features, including multi-level features (connecting features by some operations) according to pre-defined rules inspired by real-world scenarios. The pre-defined rules include the interaction with dates, coordinates, age, and address. Another interesting example is the FETCH (Li et al., 2022) that trains a single neural network to predict correct feature transformations for any tabular dataset, allowing us to accumulate knowledge about the most useful feature-dependent transformations.

### 2.3 LARGE LANGUAGE MODELS

LLMs are models intended for understanding, interpreting, and generating human-like texts (Touvron et al., 2023; Team et al., 2024). In this paper, we use the family of ChatGPT models developed by OpenAI, which are trained to perform human-like conversations and assist in various tasks. It is a relatives model of InstructGPT (Ouyang et al., 2022), designed to follow instructions in prompts and provide detailed answers. These models typically obtain state-of-the-art in the field of LLM models, so we decided to use them in our framework.

LLMs show significant cross-domain knowledge capabilities, i.e., they can transfer and apply knowledge across different domains or subject areas and solve complex problems in various fields. Knowledge can be tested by examinations (Newton & Xiromeriti, 2024). LLMs are trained on huge amounts of data spanning multiple domains, allowing them to develop broad knowledge that can be applied to various tasks and topics. According to different estimates, the size of a training sample starts from 600 GB and is obtained from multiple Internet sources (including Wikipedia). Trillions of LLM parameters (Allen-Zhu & Li, 2024) allow for structured summarization of global Internet

information, therefore it is of great interest to use LLM in the context of feature generation, because the knowledge of a data analyst in the domain-specific data domain will most often be lower than LLM.

An advanced prompting technique LLMs use to increase their ability to reason and solve problems is the Chain of thought (CoT). It motivates LLMs to decompose complex problems into intermediate steps by mimicking human reasoning processes. CoT helps LLMs solve complex problems (Feng et al., 2023). It involves asking the model to "think step by step" when answering questions or solving problems. This technique uses the model's general knowledge to improve its performance on tasks that require logic, computation, and decision-making. We add instructions like "Describe your logic and reasoning" to the prompt. Further in the paper, we ask LLMs to generate useful domain information to generate features based on it, too.

### 2.4 LLM ACTOR VS. LLM CRITIC

Industry practice and research (Gou et al., 2024) demonstrate that if we verify the answers of an LLM with another LLM by assigning the conditional roles of "actor" and "critic" in advance, i.e., a model that solves the problem and a model that catches errors in the solution of the first model, then we can improve the quality of text generation, find mistakes in advance, and improve the quality metrics in different kinds of tasks. Further in the paper, we propose to use LLM not only for feature generation but also for error catching and feature correction.

### 2.5 INTERPRETABILITY

Interpretability of features is important in the context of machine learning tasks. It is often necessary not only for the quality of the final feature pipeline but also for understanding its background and possible values to build a stable implementation. Most existing non-LLM approaches (except featuretools) do not provide proper interpretability of features, so they are selected based on optimized oversamples to improve the final model's prediction quality without caring about possible degradation of the scoring over time. This is why the LLM approach is attractive, as it can describe the reason for generating a particular feature, not only to generate the maximum number of features useful for the quality of the model (to be shown later in the paper). In addition, LLM models can be attempted to be interpreted (Singh et al., 2024) by delving deeper into the causes of generation

## 3 PROPOSED APPROACH

The proposed LLM2Features framework is shown in Fig. 1. It contains three main parts.

The first one is the **prompt generation**. The human inputs a table (training set) in pandas.DataFrame (Wes McKinney, 2010) format, and the prompt is generated using our specially developed pattern (Table 1). We use an example of feature generation and an example of LLM response for the LLM to follow the instructions more clearly. It is an example of in-context learning where the pre-trained model prior knowledge to generalize from limited task-specific data. (Parnami & Lee, 2022). The following statistics are collected from the human data: column information (data types), number of omitted values in columns, random sampling of records, distribution of values in features, and correlation in data for numerical features. Also, we request LLMs to check their answers, write all necessary information for generating further features in the [DOMAIN] section, and the features themselves in the [FEATURES] section, which increases the total number of instruction followings when generating features

**Processing the results**. The features generated using our prompt by an LLM, such as GPT-4o, are tested for validity. For example, the whole feature is removed in case of syntax errors. If the feature uses a target, it is removed. The features are typically generated by calling special libraries, such as Python standard modules or numpy (Harris et al., 2020), pandas, and geopy. If these packages are not installed, the code check unit will drop the generated feature. Next, the generated features from the training and testing set are computed for each example. In case of any error, the feature is not added. If less than three correct features are generated, the request to GPT-4o is sent again. These generation tasks **can also be effectively performed by an LLM-critic** checking the generated response. The

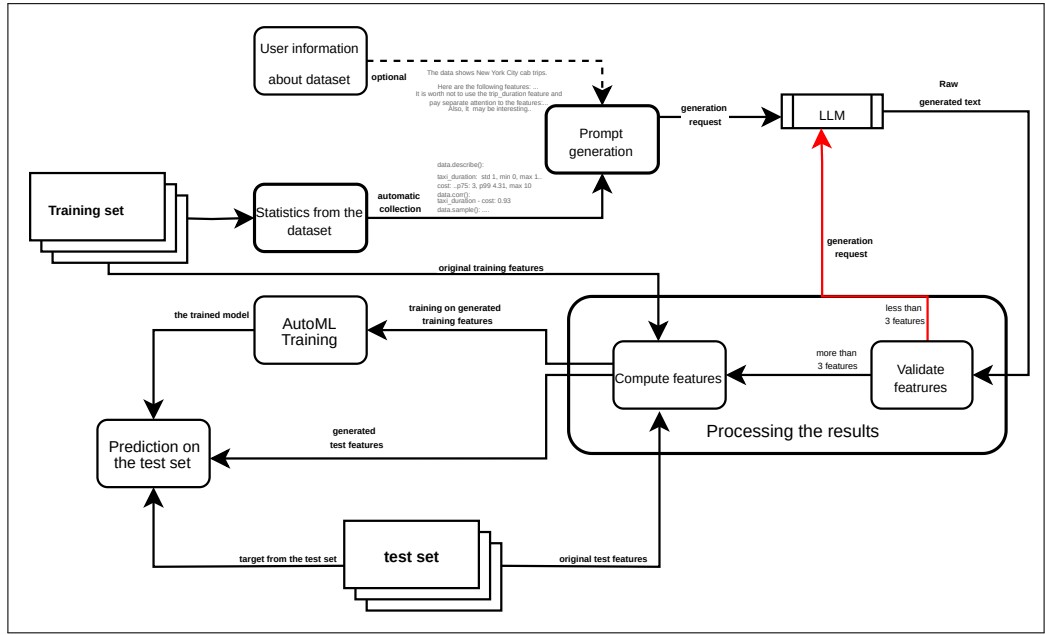

Figure 1: The proposed LLM2Features pipeline

Table 1: The prompt for our LLM-based interpretable feature generation

The dataset is loaded into the 'df' variable in pandas.DataFrame format and is available for any manipulation.
All information is taken from the dataset using the pandas library. Information about the number of omissions and
column types (data.info()):
{df.info()} 'df.sample()' (data sample): {df.sample()} 'df.corr()' (correlation for num. features):{df.corr()}
'Useful information about the data:' { **human desc of DATA (OPTIONAL)**}
'df.describe() (statistics in the features )'
{ df.describe()} This code generates additional columns based on data information, feature names, and
other useful information.
The code is posted as the best example of feature generation by LLM with extensive experience in researching data and
creating useful features The generated features are useful for solving the
{'classification' if class else 'regression'} problem using gradient boosting algorithm LightGBM
(therefore, feature generation based on scale changes or feature combinations does not make sense).
The target variable in the data is '{targetname}', the quality metric is {'ROC-AUC' if class else 'RMSE'}.
The generated features bring new logical information to the real-world data, useful for solving the problem.
Some approaches used for generation:
*. Type transformations. For example: from a numeric feature to make several categorical features
*. Creating flags. For example: putting a true flag when some conditions are fulfilled in one record,
*. Discretization. For example: divide a numerical feature into intervals and assign each interval a number
*. Complete deletion of the feature. In the case of a small number of records, it may help not to overfit
*. Changing the feature values according to the condition.
For example: replacing erroneous values with the most appropriate ones.
*. Any useful transformation based on knowledge about the real world
The information is generated in [DOMAIN]: a desc of useful information about the data and [FEATURES]:
where the Python blocks begin.
Feature generation example: {PYTHON FORMAT BLOCK}
Each block was checked for possible errors: the absence of a feature in the table and the correctness of the syntax
[LLM ANSWER] Following the proposed work format, first [DOMAIN] describing all the necessary information for
the data work is written out, followed by [FEATURES]: with the generated features in Python block format.
[DOMAIN]:

o1-preview was taken as a critic because the authors state that it is better at reason and is capable of deep thought compared to previous versions of the ChatGPT.

For simple mistakes (e.g., syntax errors), it is possible to use **ast** library[1]. For serious logical errors in features (using a feature that leaks test data, using incorrect values within a feature, etc.), the LLM-critic is used (see a prompt in Table 2)

Table 2: The prompt of our LLM-based critic

You are a LLM-critic that receives as input the output of another LLM model.
You need to fix syntax and logic errors in Python code in order to improve training quality and prediction metrics on the {**TARGET**} feature when training LightGBM on the {**CLASSIFICATION/REGRESSION**} task.
The [INIT PROMPT] will be passed first, which is the OTHER LLM prompt that was used to generate the features.
Next, the OTHER LLM response will be transmitted after the $<$ ACTOR LLM ANSWER $>$ token.
Generating start after $<$LLM CRITIC ANSWER$>$.
You should write strictly in [ERROR DESCRIPTION] format all errors that were made in the features,
including in the calculation logic or with leaked test data from the target [FIX] Python code in blocks.
Example:
[INIT PROMPT]
{**PROMPT FROM Table** 1}
$<$ ACTOR LLM ANSWER $>$
[DOMAIN] Count features 1 + 1 and remove features df['Name'] because this will come in handy
for predicting the target.
[FEATURES]
```python
# Feature: Adding two integer features
# Usefulness: This feature is needed for predicting the target.
# Input samples: 'Number_1': [1, 0, 3], 'Number_2': [0, 2, 1], 'Number_3': [4, 5, -1]
df['Sum_Number_1_and_2'] = df['Number_1'] + df['Number_2'] + df['Number_3']
```
```python
# Feature: Removing the 'Name' feature
# Usefulness: This feature does not affect targeting
df.drop(columns=['Name'])
```
$<$LLM CRITIC ANSWER$>$
[ERROR DESCRIPTION]
The feature 'Sum_Number_1_and_2' should consist of the sums of the two
integer columns 'Number_1' and 'Number_2'. Since 'Number_3' is added to it, this sign contains an error
The 'Name' sign was not deleted because the inplace=True argument is missing.
[FIX]
```python
# Feature: Adds two integer features
# Usefulness: This feature is needed for predicting the target.
# Input samples: 'Number_1': [1, 0, 3], 'Number_2': [0, 2, 1]
df['Sum_Number_1_and_2'] = df['Number_1'] + df['Number_2']
```
```python
# Feature: Removing the 'Name' feature
# Usefulness: This feature does not affect targeting
df.drop(columns=['Name'], inplace=True)
```
[INIT PROMPT]:

The final part of our pipeline is the **Auto ML training**. Appropriate AutoML framework, such as LightAutoML (Vakhrushev et al., 2021), is trained on the generated features and predicts a test sample.

---

[1]https://docs.python.org/3/library/ast.html

Table 3: Datasets for experimental study

| Task | Dataset | Target | No. Rows (train) | No. Rows (test) | No. features |
|------|---------|--------|------------------|-----------------|--------------|
| binary classification | Titanic Cukierski (2012) | A titanic passenger's survival flag | 534 | 214 | 11 |
| | Credit-g Hofmann (2014) | A customer credit risk flag | 750 | 250 | 21 |
| | Diabetes Kaggle (2020) | A flag for the presence of diabetes | 576 | 192 | 9 |
| regression | California Housing Price Nugent (2017) | Forecasting housing prices | 12384 | 4953 | 9 |
| | NYC Taxi Duration Risdal (2017) | A ride duration of taxi trips | 100000 | 100000 | 10 |
| | Mental Health ASHFAQ (2024) | A mental state of students | **65** | **22** | 19 |

## 4 EXPERIMENTS

### 4.1 EXPERIMENTAL SETTINGS

The proposed LLM2Features approach is implemented in two supported scenarios: 1) with a description of data, including information about domain and attributes, and 2) without data description, where features are generated only based on statistics from the dataset. We use two state-of-the-art LLMs from OpenAI, namely, GPT-4, GPT-4o (Achiam et al., 2023), GPT-o1-preview(OpenAI, 2024), that can better follow rather complex prompt (Table 1).

In addition to using initial features, we compare our pipeline with several state-of-the-art feature generation techniques, such as **OpenFE** (Zhang et al., 2023), **AutoFeat** (Horn et al., 2020) and **Featuretools** (Kanter & Veeramachaneni, 2015). Moreover, we used an official **LightAutoML Pipeline**[2] (Vakhrushev et al., 2021), which is an example of a classic approach for AutoML frameworks. It simply encodes categorical features, transforms some data, and boosts selectors. Finally, we implemented the state-of-the-art LLM-based feature generator with the domain knowledge about the dataset, **CAAFE** (Hollmann et al., 2024) with GPT-4 and GPT-4o. Special preprocessing was only applied to the data if the method for feature generation did not work without accurate type conversion, omission, and anomaly correction. Omissions were filled with median (statistics were counted with the condition of not allowing leakage of test data), feature types were corrected by the meaning of the feature and the needs of specific algorithms for feature generation (for example, the basic implementation of featuretools requires type conversion using woodwork [3]).

The proposed approach is implemented in two settings:

1. **LLM2Features with human input**: with domain information, attributes, and data
2. **LLM2Features without human input**: without domain information, attributes, and data (attributes are generated only based on statistics from the dataset). It is the most suitable for application with AutoML frameworks.

In our experiments, we examine several traditional datasets for binary classification and regression tasks (Table 3) that are widely used in various papers (Hollmann et al., 2024; Katz et al., 2016; Kaul et al., 2017; Li et al., 2022). We use the modern AutoML framework, LightAutoML (Vakhrushev et al., 2021) v0.3.8.1, to train classification and regression models, which has recently won the Kaggle's AutoML Grand Prix 2024. We compute traditional metrics, namely, F1-score and ROC-AUC (Area Under the ROC Curve) for classification and RMSE (Root Mean Squared Error) and MAPE (Mean Absolute Percentage Error) for single regression. Moreover, we estimate the performance

---

[2] https://colab.research.google.com/github/AILab-MLTools/LightAutoML/blob/master/examples/tutorials/Tutorial_6_custom_pipeline.ipynb

[3] https://woodwork.alteryx.com/en/stable/

Table 4: Experimental results for classification tasks

| Dataset | Method | Total (sec.) | Init (sec.) | Fit (sec.) | Predict (sec.) | F1 | ROC AUC | H.I. |
|---|---|---|---|---|---|---|---|---|
| Titanic | Initial features | 168.83 | 0.042 | 168.726 | 0.062 | 0.773 | 0.869 | - |
| | LightAutoML pipeline | 24.959 | 0.008 | 24.888 | 0.062 | 0.748 | 0.856 | - |
| | OpenFE | 296.853 | 0.086 | 295.971 | 0.796 | 0.752 | 0.856 | 2 |
| | featuretools | 225.63 | 0.06 | 225.334 | 0.235 | 0.762 | 0.868 | 2 |
| | autofeat | 370.676 | 0.032 | 370.491 | 0.152 | 0.756 | 0.871 | 2 |
| | CAAFE (GPT-4) | 170.222 | 0.046 | 169.957 | 0.22 | 0.763 | 0.878 | 1 |
| | CAAFE (GPT-4o) | 566.649 | 0.106 | 566.266 | 0.277 | 0.800 | 0.884 | 1 |
| | **LLM2Features with description (GPT-4)** | 252.639 | 0.106 | 252.345 | 0.189 | 0.795 | 0.885 | 1 |
| | **LLM2Features with description (GPT-4o)** | 379.663 | 0.067 | 379.355 | 0.240 | **0.81** | 0.887 | 1 |
| | **LLM2Features without description (GPT-4o)** | 380.254 | 0.071 | 380.120 | 0.063 | 0.761 | 0.868 | **0** |
| Credit-g | Initial features | 276.138 | 0.114 | 275.789 | 0.235 | 0.844 | 0.794 | - |
| | LightAutoML pipeline | 55.994 | 0.009 | 55.932 | 0.054 | 0.843 | 0.756 | - |
| | OpenFE | 753.897 | 0.166 | 751.444 | 2.287 | 0.844 | 0.793 | 2 |
| | featuretools | 666.235 | 0.103 | 665.664 | 0.468 | 0.844 | 0.800 | 2 |
| | autofeat | 1826.556 | 0.088 | 1826.4 | 0.068 | 0.841 | 0.779 | 2 |
| | CAAFE (GPT-4) | 326.308 | 0.047 | 325.982 | 0.279 | 0.85 | 0.799 | 1 |
| | CAAFE (GPT-4o) | 481.956 | 0.167 | 481.528 | 0.260 | 0.851 | 0.778 | 1 |
| | **LLM2Features with description (GPT-4)** | 298.599 | 0.058 | 298.218 | 0.323 | **0.852** | 0.795 | 1 |
| | **LLM2Features with description (GPT-4o)** | 471.436 | 0.255 | 470.43 | 0.75 | 0.847 | **0.801** | 1 |
| | **LLM2Features without description (GPT-4o)** | 450.747 | 0.098 | 450.346 | 0.303 | 0.84 | 0.797 | **0** |
| Diabetes | Initial features | 419.682 | 0.132 | 419.427 | 0.123 | 0.662 | 0.803 | - |
| | LightAutoML pipeline | 32.218 | 0.011 | 32.157 | 0.05 | 0.559 | 0.796 | - |
| | OpenFE | 665.415 | 0.097 | 663.615 | 1.703 | 0.652 | 0.797 | 2 |
| | featuretools | 444.322 | 0.175 | 444.077 | 0.071 | 0.662 | 0.803 | 2 |
| | autofeat | 453.782 | 0.023 | 453.642 | 0.117 | 0.623 | 0.803 | 2 |
| | CAAFE (GPT-4) | 289.287 | 0.073 | 289.103 | 0.111 | 0.627 | 0.796 | 1 |
| | CAAFE (GPT-4o) | 442.723 | 0.207 | 442.364 | 0.152 | 0.647 | 0.803 | 1 |
| | **LLM2Features with description (GPT-4)** | 245.668 | 0.071 | 245.47 | 0.127 | 0.662 | 0.803 | 1 |
| | **LLM2Features with description (GPT-4o)** | 431.542 | 0.082 | 431.206 | 0.253 | **0.686** | **0.813** | 1 |
| | **LLM2Features without description (GPT-4o)** | 408.503 | 0.145 | 408.158 | 0.2 | 0.63 | 0.8044 | **0** |

of the following stages in a typical pipeline on a PC with an Intel Xeon processor with two virtual CPUs and 12.5 GB RAM:

- **Init (sec.)**, time to initialize all necessary methods.
- **Fit (sec.)**, time to perform fit speed, which contains both running the feature generation and training the model. As requests to the LLM do not exceed 30 sec., we add 30 sec for LLM-based feature generation.
- **Predict (sec.)**, time to perform prediction for the complete test set, including computation of generated features.
- **Total (sec.)**, the total running time of AutoML that is a sum of Init, Fit, and Predict.

## 4.2 NUMERICAL RESULTS

The results of our experiments for classification and regression tasks are shown in Table 4 and Table 5, respectively. Suppose the proposed LLM2Features is operated without additional information from humans, i.e., using only statistics from the dataset. In that case, obtaining significantly better metrics than traditional feature-generation techniques for all datasets is possible. However, using additional domain knowledge in the CAAFE (Hollmann et al., 2024), a previous application of LLM for feature engineering, can increase the overall accuracy. Nevertheless, the proposed approach with additional domain information performed better than the CAAFE baseline (Hollmann et al., 2024) on all binary classification and regression metrics datasets. It is also worth noting that the proposed approach is faster than any non-LLM approach, despite a rough estimate of 30 seconds for LLM generation.

Table 5: Experimental results for regression tasks

| Dataset | Method | Total (sec.) | Init (sec.) | Fit (sec.) | Predict (sec.) | RMSE | MAPE | H.I. |
|---------|--------|--------------|-------------|------------|----------------|------|------|------|
| Taxi | Initial features | 737.675 | 0.06 | 735.215 | 2.4 | 3127.033 | 1.172 | - |
| | LightAutoML pipeline | 175.305 | 0.004 | 172.201 | 3.1 | 3156.163 | 1.528 | - |
| | OpenFE | 1235.321 | 0.751 | 1231.416 | 3.153 | 3097.566 | 1.357 | 2 |
| | featuretools | 772.226 | 2.018 | 767.898 | 2.311 | 2904.571 | 0.818 | 2 |
| | autofeat | 731.505 | 0.25 | 730.321 | 0.934 | 3149.696 | 1.495 | 2 |
| | CAAFE (GPT-4) | 866.167 | 0.035 | 863.274 | 2.859 | 2243.413 | 0.839 | 1 |
| | CAAFE (GPT-4o) | 773.673 | 0.122 | 771.722 | 1.830 | 3138.281 | 1.336 | 1 |
| | **LLM2Features with description (GPT-4)** | 869.667 | 0.058 | 867.59 | 2.019 | 2174.115 | 0.67 | 1 |
| | **LLM2Features with description (GPT-4o)** | 1005.72 | 0.036 | 1003.548 | 2.137 | **2157.126** | **0.664** | 1 |
| | **LLM2Features without description (GPT-4o)** | 683.106 | 0.203 | 682.903 | 3.368 | 2571.521 | 0.941 | **0** |
| House | Initial features | 713.203 | 0.037 | 713.959 | 9.206 | 0.445 | 0.164 | - |
| | LightAutoML pipeline | 238.901 | 0.003 | 235.363 | 3.534 | 0.447 | 0.168 | - |
| | OpenFE | 1550.089 | 0.104 | 1543.483 | 6.502 | 0.443 | 0.164 | 2 |
| | featuretools | 746.134 | 0.106 | 739.488 | 6.54 | 0.445 | 0.164 | 2 |
| | autofeat | 1007.544 | 0.021 | 1001.955 | 5.568 | 0.454 | 0.165 | 2 |
| | CAAFE (GPT-4) | 752.779 | 0.038 | 746.457 | 6.284 | 0.451 | 0.165 | 1 |
| | CAAFE (GPT-4o) | 612.440 | 0.065 | 607.093 | 5.282 | 0.447 | 0.164 | 1 |
| | **LLM2Features with description (GPT-4)** | 815.578 | 0.038 | 809.493 | 6.046 | 0.448 | 0.164 | 1 |
| | **LLM2Features with description (GPT-4o)** | 774.347 | 0.125 | 767.185 | 7.037 | **0.442** | **0.163** | 1 |
| | **LLM2Features without description (GPT-4o)** | 617.345 | 0.185 | 615.332 | 1.829 | 0.467 | 0.172 | **0** |

Table 6: Regression datasets with a minimum number of records, comparison of statistical methods, and o1-preview (chosen by metrics over other LLM approaches)

| Dataset | Method | Total (sec.) | Init (sec.) | Fit (sec.) | Predict (sec.) | RMSE | MAPE | H.I. |
|---------|--------|--------------|-------------|------------|----------------|------|------|------|
| Mental Health | Initial features | 380.593 | 0.075 | 380.309 | 0.209 | 1.187 | 7e+14 | - |
| | LightAutoML pipeline | 18.765 | 0.004 | 18.748 | 0.013 | 1.212 | 7.4e+14 | - |
| | featuretools | 225.101 | 0.048 | 224.394 | 0.658 | 1.142 | 6.5e+14 | |
| | autofeat | 590.479 | 0.12 | 590.168 | 0.191 | 1.175 | 6.92e+14 | 2 |
| | **LLM2Features without description (GPT-o1-preview)** | 448.270 | 0.118 | 447.586 | 0.566 | **0.997** | **5e+14** | **0** |

### 4.3 QUALITATIVE EXAMPLES

A qualitative example of the feature generated by our approach for the Diabetes dataset is shown in Table 7. Here, LLM demonstrates an understanding of the correlation between weight and diabetes risk in a way that may not be obvious to a person without a medical background. The generated feature can be a great starting point for further deeper analysis.

Table 8 shows our LLM-based critic's understanding of the errors associated with generating features with known test data leakage or LLM hallucination and correct error values in the feature.

In Table 7, the features from our approach not only improve the quality of the AutoML model but also can be easily interpreted for further manual feature engineering or creating additional features using classical feature generation techniques.

Finally, in Table 9 (Appendix A), we can see a clear division of the LLM response into zones, the first one is where task formulation takes place and the second one generates syntactically correct features that can be immediately used to run the generation task. Curiously, the LLM presented in Table 9 does not generate a description based on Python-generated features but first invents a feature, describes it, and only encodes it into Python code. This is due to the nature of the model's design, which essentially consists of sequential token generation. Based on the features obtained, further analysis can be completed. For example, examine the "Name" feature in more detail for possible other insights or examine the "Cabin" feature that may affect survival rates.

Table 7: Example of features generated by our approach with automatic LLM-based interpretations

| Dataset | Sample feature | Interpretation by LLM |
|---------|----------------|----------------------|
| Titanic | df['Title'] = df['Name'].str.extract(' ([A-Za-z]+)', expand=False) | Titles can provide information on social status and gender. How could an analyst create it in the real world: by extracting the title (e.g., Mr., Mrs., Miss) from the attribute "name". |
| Diabetes | df['BMICategory'] = pd.cut(df['BMI'], bins=[0, 18.5, 24.9, 29.9, 50], labels=['Underweight', 'Normal', 'Overweight', 'Obese']) # Input samples: 'BMI': [21.8, 25.3, 30.5] | BMICategory: Categorize BMI into Underweight, Normal, Overweight, and Obese. Usefulness: BMI categories provide insight into the patient's weight status, which is crucial for diabetes risk. |
| Credit G | df['HighRisk'] = df['HighRisk'].apply( lambda x: 1 if $x > 3$ else 0) # Input samples: 'HighRisk': [2.0, 3.0, 4.0] | Useful for class prediction as high installment commitments might indicate financial strain |
| Mental Health | ddf['sleep_deprived'] = df['average_sleep'].isin(['<5 hrs', '5-6 hrs']) | Sleep deprivation can affect mental health and academic performance, impacting survival. |
| Housing Price | df['AgeCategory'] = pd.cut(df['HouseAge'], bins=[0, 10, 30, 50, 100], labels=['New', 'Medium', 'Old', 'Very Old']) | Categorizing house age into newer and older can help capture nonlinear effects on house value, as newer homes might have higher values due to less wear and more modern amenities. |
| Taxi | df['is_rush_hour'] = df['pickup_hour'].isin([7, 8, 9, 16, 17, 18]) | Flags trips during peak traffic times, which can affect trip duration. |

## 5 CONCLUSION

This paper proposes a novel LLM-based automatic feature generation approach for AutoML with tabular data (Fig. 1). We experimentally proved that the proposed approach lead to better performance (Tables 4, 5) when compared to conventional state-of-the-art feature generators. It is important to emphasize that we improved metrics not for outdated ML models but using the contemporary LightAutoML framework (Vakhrushev et al., 2021). Moreover, our method generates interpretable features for regression and classification tasks based only on feature names and data statistics without an additional description (Table 7). We are the first to present this extremely important property for LLM-based auto feature generation. As a result, our LLM2Features pipeline can be integrated into an arbitrary AutoML pipeline for complete no-code ML. Also, the described approach can be used for preliminary data analysis. The LLM generates valuable features for AutoML and allows humans to explore unknown data in greater detail. The source code of our experiments will be made publicly available[4].

In future, it is necessary to extend our LLM2Features approach, e.g., use more examples of feature generation in the prompt. It is also important to highlight that our current approach sends all requests to the LLM on a one-time request. Hence, it is possible to increase the accuracy by sending requests to adjust features with their importance for the obtained AutoML model. Finally, applying our

---

[4]https://anonymous.4open.science/r/LLM2Features-1B28

Table 8: Example of our LLM-based critic that fixes the issues with features generated by a LLM

| Feature | Problem | [ERROR DESCRIPTION] | [FIX] |
|---|---|---|---|
| df['dropoff_datetime'] = pd.to_datetime( df['dropoff_datetime'] ) | This feature is from the Taxi (Risdal, 2017), where we need to predict the length of a trip. The dropoff_datetime feature is not available to us, it was mentioned in the data description | The dropoff_datetime feature defines the end time of the trip, therefore feature can not be included in the test data for the forecast. The feature should be deleted | df.drop(columns = ['dropoff_datetime'], inplace=True) |
| df['sleep_deprived'] = df['average_sleep'].isin( ['> 5 hrs', '5-6 hrs'])) | There is a typographical error. The values in the 'average_sleep' feature are only <5 (ASH-FAQ, 2024) | In the definition of a feature, a typing error has been committed | df['sleep_deprived'] = df['average_sleep'].isin( ['<5 hrs', '5-6 hrs'])) |
| df['duration_group'] = pd.cut(df['duration'], bins=[0, 5, 10, 20, 30, 40, 50], labels=['very_short', 'short', 'medium', 'long', 'very_long', 'extra_long']) | Not appropriate duration for the described loan groups (Hofmann, 2014). | The bins for the feature do not fit well with the specified groups | df['duration_group'] = pd.cut(df['duration'], bins=[0, 12, 24, 36, 48, 60, 120], labels=['very_short', 'short', 'medium', 'long', 'very_long', 'extra_long']) |

models with the publicly-available LLMs, which may need to simplify instructions in our prompt (Table 1).

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

# A    ADDITIONAL QUALITATIVE RESULTS

Table 9: ChatGPT-4o Answer to the Titanic dataset prompt (without data description), subsample of generated features

[DOMAIN]:
We are analyzing the Titanic dataset to predict passenger survival ('Survived') using a gradient boosting algorithm, with ROC-AUC as the evaluation metric. The dataset contains various features such as socioeconomic status (Pclass), personal information (Name, Sex, Age), family
relations (SibSp, Parch), ticket details, fare, cabin assignment, and port of embarkation (Embarked). Our objective
is to generate new features that introduce meaningful real-world insights to enhance model performance.
We will focus on transformations that capture social status, family connections, passenger demographics,
and other relevant information.
[FEATURES]:
``` python
# Feature: Extracting Title from passenger's name
# Usefulness: Titles can indicate social status and marital status, which may correlate with survival rates.
# Input samples:
# 'Name': ['Braund, Mr. Owen Harris', 'Heikkinen, Miss. Laina', 'Allen, Master. William Henry']
df['Title'] = df['Name'].str.extract(' ([A-Za-z]+)', expand=False)
# Feature: Simplifying titles into common categories
# Usefulness: Grouping rare titles reduces noise and captures essential social distinctions.
# Input samples:
# 'Title': ['Mr', 'Miss', 'Master', 'Dr', 'Lady']
title_mapping = {'Mr': 'Mr', 'Miss': 'Miss', 'Mrs': 'Mrs',
'Master': 'Master', 'Dr': 'Officer', 'Rev': 'Officer', 'Col': 'Officer',
'Major': 'Officer', 'Mlle': 'Miss', 'Countess': 'Royalty', 'Ms': 'Mrs',
'Lady': 'Royalty', 'Jonkheer': 'Royalty', 'Don': 'Royalty', 'Dona': 'Royalty',
'Mme': 'Mrs', 'Capt': 'Officer', 'Sir': 'Royalty' }
df['Title'] = df['Title'].map(title_mapping)
# Feature: Creating FamilySize from SibSp and Parch
# Usefulness: Family size may affect survival chances; larger families might have different survival rates.
# Input samples:
# 'SibSp': [1, 0, 3], 'Parch': [0, 2, 1]
df['FamilySize'] = df['SibSp'] + df['Parch'] + 1
# Feature: Creating IsAlone flag based on FamilySize
# Usefulness: Passengers traveling alone may have different survival probabilities compared to those with family.
# Input samples:
# 'FamilySize': [1, 4, 2]
df['IsAlone'] = 0
df.loc[df['FamilySize'] == 1, 'IsAlone'] = 1
# Feature: Discretizing Age into age groups
# Usefulness: Age groups can capture nonlinear relationships between age and survival.
# Input samples:
# 'Age': [22, 38, 4]
df['AgeGroup'] = pd.cut(df['Age'], bins=[0, 12, 18, 35, 60, 80], labels=[1, 2, 3, 4, 5])
# Feature: Extracting Deck information from Cabin
# Usefulness: Deck level may influence survival due to proximity to lifeboats.
# Input samples:
# 'Cabin': ['C85', NaN, 'E46']
df['Deck'] = df['Cabin'].str[0]
df['Deck'].fillna('U', inplace=True)
# Mapping Deck to numerical values
# Usefulness: Converts deck letters to numerical codes for modeling.
# Input samples:
# 'Deck': ['C', 'U', 'E']
deck_mapping = 'A':1, 'B':2, 'C':3, 'D':4, 'E':5, 'F':6, 'G':7, 'T':8, 'U':0
df['Deck'] = df['Deck'].map(deck_mapping)
# Dropping uninformative or redundant features
# Explanation: 'Ticket' and 'Cabin' are dropped due to high missing values and redundancy after extracting 'Deck'.
df.drop(columns=['Ticket', 'Cabin'], inplace=True)
# Feature: Binning Fare into discrete categories
# Usefulness: Fare can have nonlinear effects; binning captures these patterns.
# Input samples:
# 'Fare': [7.25, 71.2833, 8.05]
df['FareBand'] = pd.qcut(df['Fare'], 4, labels=[1, 2, 3, 4])
```

