# OpenReview forum: "LLM2Features: Large Language Models in Interpretable Feature Generation for AutoML with Tabular Data"
_ICLR.cc/2025/Conference — ICLR 2025 Conference Withdrawn Submission_

### Official Review · Reviewer_azwo · 2024-10-22

**Soundness:** 1
**Presentation:** 1
**Contribution:** 1
**Rating:** 1
**Confidence:** 5

**Summary:**

The paper proposes LLM2Features, an automated feature engineering method pipeline that does not need human input to perform feature engineering with LLMs. Thus making it easier to use than previous work in AutoML pipelines. The paper also claims that the resulting features are more interpretable than traditional automated feature engineering methods.

The paper extends prior work on automated feature engineering with LLMs by adding an actor-critic verification fallback. Moreover, the paper adds statistical information such as `df.describe()` to the prompt (together with or instead of human descriptions). Furthermore, LLM2Features also contains an integrated AutoML system to obtain the final predictions.

**Strengths:**

The paper proposes three interesting additions to a typical automated feature engineering workflow with LLMs, which are worthy of future investigation: integrating feature engineering LLMs into AutoML pipelines, the concept of a critic for generated features, and adding statistical information to the prompt.

**Weaknesses:**

# Major
## State of Experiments
The experiments do not allow for any general claims in the paper. In short, these experimental design is severely limited.

The paper disregarded many standards for benchmarking tabular machine learning methods and feature engineering. Prior work cited in this paper has done this much better, but the author may also consider work like the AutoML Benchmark (https://arxiv.org/abs/2207.12560) to obtain a complete picture.

To provide a short list of failures sorted by impact on the results:

* The paper does not contain any memorization tests for the tabular dataset used in the evaluation. As a result, it is impossible to determine from the paper if the results come from the LLM memorizing feature engineering or capitalizing on its world knowledge. See https://arxiv.org/abs/2404.06209 for an extended discussion. Furthermore, several of the datasets used in this paper have already been shown to be memorized by GPT-4 (the LLM used in this paper).
* The paper uses a small set of datasets typically used as toy examples. The results for six datasets do not allow us to make any general claims. Furthermore, the references for these datasets are not original; all are sourced from Kaggle. See the UCI Machine Learning repository or OpenML.org for details on the trace and original citations. I recommend using the curated set of datasets from the AutoML Benchmark or similar suites from prior work with at least 25 datasets.
* The paper did not repeat any experiments (e.g., with cross-validation) to assess the variability of the performance. I recommend the standard practice of 10-fold cross-validation (with stratification for classification).
* The papers report two metrics but do not specify which metrics the AutoML system and the automated feature engineer methods were tuned for. By default, neither of these tunes performs for the F1 score.
* The paper uses temporal feature engineering (like featuretools) compared to non-temporal (like OpenFE and AutoFE) while working on both temporal and non-temporal datasets.
* The time given to AutoML might not be enough to fit proper models (that might not even need feature engineering). Typically, the field gives AutoML 1 hour to 4 hours for AutoML applications.


# Minor & Other
## Questionable Contribution
The introduction claims the paper's main contribution is using LLMs for feature engineering without human descriptions. Yet, the results show that human descriptions are crucial for performance. Thus, if we look at the prompt template, the main contribution is adding `df.describe` and `df.corr` to a CAAFE-like prompt while optionally removing the human description without a positive effect. On the method side, the improvement is adding actor-critic behavior while removing CAAFE's feedback loop, thus trading one form of feedback for another without exploring this in more detail. Finally, while this is mentioned a lot in the introduction, I do not believe this work adds anything new to interpretability for automated feature engineering.

## Problematic Clarity
The writing of the paper can be improved by focusing on paragraph-level structure and ensuring that results or references back up claims. The related work section contains content that is not related, part of the method, or background material. Moreover, the results are confusingly presented. The paper shows different metrics without explaining the necessary details or the need to show these, i.e., which claims the metrics in the tables try to support.

To highlight some types:
* Line 57 "AutopML"
* Line 77 missing closing parentheses
* Line 295 Missing white space

For example, consider the following claims:
* Line 100 "These models typically obtain state-of-the-art in the field of LLM models, so we decided to use them in our framework." - consider citing https://lmarena.ai/
* Line 108 "because the knowledge of a data analyst in the domain-specific data domain will most often be lower than LLM.", Do we have evidence that LLMs are competent data scientists? The best I know is something like DSBench, which shows quite the opposite: https://arxiv.org/abs/2409.07703

## Problematic Interpretability Claims
I disagree with the notion propagated in the introduction and related work that AutoFE or OpenFE produce uninterpretable new features. They both only apply very easy-to-use operators (add, groupby, …) to the data. The outcome is then sorted, for example, by feature importance, and practitioners can clearly see how new features were generated.

I also failed to follow how featuretools, which use Deep Feature synthesis, are considered to be more interpretable because they also use feature primitives. Moreover, they focus on relational and temporal data, which neither AutoFE nor OpenFE focus on.

The argument made in 2.5 (Line 131) is that OpenFE and AutoFE are not interpretable because they validate on validation data whether a new feature is good or not. Meanwhile, LLMs return a (hallucinated) response with potentially made-up reasons that do not need to lead to better performance or understandable features.

These arguments and claims made for interpretability are also only presented in the introduction and related work and are almost absent from the results, except for the explanations produced by the LLM, which are similar to explanations produced by CAAFE. Likewise, these explanations could just be the result of memorization.

**Questions:**

I do not understand the focus of the title, abstract, introduction, and related work on interpretability as it is not a verified novel contribution of LLM2Features. Explaining this motivation and framing it in detail would help a lot. Moreover, adding more experiments or clear case-by-case comparisons of what is not interpretable to the results would benefit such claims.

I suggest focusing on individual methodological changes (like actor-critic instead of a feedback loop) instead of designing a new pipeline or comparing CAAFE with the same pipeline.

For the experiments, I suggest the following:
* Make sure always to use memorization tests for tabular data when using LLMs.
* Consider using repeated cross-validation and more datasets (not from Kaggle but with shared splits like from OpenML).
* Optimize for the target metric in AutoML and automated feature engineering tools with a pre-defined time-limit.
* Use a set of curated datasets like benchmarking suites used in prior work.

---

### Official Review · Reviewer_xTNP · 2024-11-01

**Soundness:** 2
**Presentation:** 1
**Contribution:** 1
**Rating:** 1
**Confidence:** 4

**Summary:**

This paper introduces a technique using LLM to generate additional features to augment classification and regression on tabular data. Features are generated with statistics of the datasets and artificial prompt engineering, coupled with a feature "critic" to ensure their quality. These features are fed into LightAutoML for prediction.

**Strengths:**

1. The feature generation pipeline is simple and could be applied to any downstream prediction algorithm.
2. The pipeline still has reasonable performance without human input.

**Weaknesses:**

1. Poor paper quality: most tables and prompts are content fillers and poorly formatted. LightAutoML is used for prediction, but there is no concrete description despite repeated citations. Many errors in the text show a lack of serious proofreading.
2. Experiments are limited to a few datasets, while other LLM-related works test on benchmarks with tens of datasets. No ablation study is conducted.
3. Without error bars, it is hard to tell if LLM2Features performs better than CAAFE. Most results are very close.

**Questions:**

Major revision of the paper is needed for the standard of ICLR.

---

### Official Review · Reviewer_JFvU · 2024-11-04

**Soundness:** 2
**Presentation:** 2
**Contribution:** 2
**Rating:** 5
**Confidence:** 4

**Summary:**

In this paper, the author intends to optimize the tabular feature generation process by applying LLMs. Specifically, they introduce a novel framework called LLM2Features, which automatically extracts statistical information from data and creates new features based on this information. Benefiting from LLMs' information comprehension capabilities, the LLM2Features framework improves the clarity of new features and doesn't need additional manual input during the feature generation process.

**Strengths:**

1. LLM2Features represent a novel way of applying statistical data from datasets to assist LLMs in generating features. Many existing approaches focus on using rich external information to generate features, which can be challenging when external information, such as dataset and attribute descriptions, is sparse or unavailable. Therefore, the ability of this framework to generate interpretable and task-relevant features without relying on external information is valuable.

2. LLM2Features is not tied to any specific AutoML framework, making it relatively easy to transfer and integrate into existing AutoML tools. This flexibility broadens its applicability.

3. The paper conducts experiments across multiple datasets covering classification and regression tasks and compares the proposed method with representative baselines, demonstrating its effectiveness.

**Weaknesses:**

1. Although the authors introduce an LLM-based critic model to check generated features, it cannot fully ensure that there are no errors or data leakage issues during the feature generation process. Language models may still produce features that are not entirely relevant to the data or are logically inconsistent, especially when they lack sufficient understanding of certain domains.

2.LLM2Features applies LLMs for feature generation influenced by randomness. Different LLM responses could lead to inconsistencies in the features generated for the same dataset across different runs, resulting in varying performance. The authors should consider adding repeated experiments or cross-validation in their study to mitigate the randomness introduced by LLMs.

3. The authors could focus on presenting the LLM2Features more clearly. Specifically, in the proposed approach section, authors should provide a more detailed and transparent explanation of how the LLMs, acting as actor and critic, acquire the necessary information to generate or process new features, rather than only presenting a prompt template. Additionally, in Figure 1, most of the text representing operations like ' original training features' or 'generation request' is too small, and the font color next to "prompt generation" is too light, making it difficult to read.

**Questions:**

1. In this paper’s experiments, the datasets used seem to have no more than 30 original features. How would LLM2Features handle datasets with a larger number of original features that require generating a greater number of new features? This question is particularly relevant considering the use of o1-preview as the critic model for validation—would the cost increase significantly as the number of features grows?

2. LLM2Features relies on LLMs for feature generation, but LLMs may introduce biases or generate unreasonable features. How can the method effectively detect and correct biases or unreasonable features introduced during the generation process, especially those carried over from previous iterations?

---

### Official Review · Reviewer_umzf · 2024-11-04

**Soundness:** 3
**Presentation:** 2
**Contribution:** 2
**Rating:** 5
**Confidence:** 3

**Summary:**

The paper proposes LLM2Features, a novel feature-generation approach leveraging large language models (LLMs) to enhance the interpretability and automation of feature engineering for tabular data in AutoML systems. Unlike traditional methods that often require human input or lack interpretability, LLM2Features autonomously generates features using only dataset statistics, omitting explicit domain descriptions. It introduces a two-step "LLM actor-critic" mechanism where the LLM generates features (actor), followed by a secondary LLM that checks for logical and syntactic errors (critic). The method is validated against benchmark datasets in both classification and regression contexts, where it outperforms other feature-generation methods, particularly on interpretable features. Empirical results show improvements in both predictive accuracy and training efficiency, suggesting that LLM2Features is a viable addition to AutoML pipelines for data exploration and model interpretability.

**Strengths:**

LLM2Features represents an innovative use of LLMs for automated feature engineering, providing a unique contribution to AutoML by reducing reliance on human-supplied domain knowledge.

By using interpretable feature generation, the method allows users to understand the logical basis of generated features, a step forward for transparent ML workflows.

**Weaknesses:**

The reliance on specific LLM models (such as GPT-4) may restrict reproducibility or generalizability in environments without access to these high-capacity LLMs, impacting the wider applicability of the method.

Although interpretability is emphasized, the paper lacks a clear metric or method for quantitatively assessing the interpretability of the generated features. Providing such an assessment could make the claims more robust.

Although the critic model attempts to catch logical errors, the error-checking mechanism's accuracy and completeness are not fully validated, which could lead to overlooked or latent errors in complex datasets.

**Questions:**

Please see the weakness section.

---

### Note · Authors · 2024-11-28

**Comment:**

Dear reviewers!
Thanks for your detailed feedback and advices. As it seems we need a bit more time to carry out additional experiments and improve clarity of our presentation, we decided to withdraw this paper

**Withdrawal Confirmation:**

I have read and agree with the venue's withdrawal policy on behalf of myself and my co-authors.